# Golgi membrane protein Erd1 Is essential for recycling a subset of Golgi glycosyltransferases

Richa Sardana[1,2]*, Carolyn M Highland[1], Beth E Straight[1], Christopher F Chavez[1], J Christopher Fromme[1], Scott D Emr[1]*

[1]Department of Molecular Biology and Genetics, Weill Institute for Cell and Molecular Biology, Cornell University, Ithaca, United States; [2]Department of Molecular Medicine, Cornell University, Ithaca, United States

**Abstract** Protein glycosylation in the Golgi is a sequential process that requires proper distribution of transmembrane glycosyltransferase enzymes in the appropriate Golgi compartments. Some of the cytosolic machinery required for the steady-state localization of some Golgi enzymes are known but existing models do not explain how many of these enzymes are localized. Here, we uncover the role of an integral membrane protein in yeast, Erd1, as a key facilitator of Golgi glycosyltransferase recycling by directly interacting with both the Golgi enzymes and the cytosolic receptor, Vps74. Loss of Erd1 function results in mislocalization of Golgi enzymes to the vacuole/lysosome. We present evidence that Erd1 forms an integral part of the recycling machinery and ensures productive recycling of several early Golgi enzymes. Our work provides new insights on how the localization of Golgi glycosyltransferases is spatially and temporally regulated, and is finely tuned to the cues of Golgi maturation.

## Editor's evaluation

The authors took the critiques seriously and responded with substantial changes and enhancements. While we are still not fully convinced by the interpretations, the authors have done a thorough, rigorous piece of work that makes a valuable contribution about a topic of broad interest.

*For correspondence:
sardana@cornell.edu (RS);
sde26@cornell.edu (SDE)

**Competing interest:** The authors declare that no competing interests exist.

## Introduction

Glycosylation is the most abundant and diverse post-translational protein modification and plays a critical role in modulating protein interactions, stability, and physiological functions *Stanley, 2011*; *Schjoldager et al., 2020*. The importance of proper glycosylation is highlighted by the multi-system failures, immune dysfunction, and many inherited disorders of glycosylation arising due to the loss of activity or localization of Golgi enzymes *Reily et al., 2019* ; *Ng and Freeze, 2018*; *Rodrigues et al., 2018*. The complex structure of glycosylation modifications is generated by the sequential action of specific sets of membrane-embedded glycosyltransferases localized to distinct Golgi cisternae *Schjoldager et al., 2020*; *Colley, 1997*; *Orlean, 2012*. Maintenance of these enzymes in the appropriate Golgi compartments therefore allows accurate protein glycosylation *Moremen et al., 2012*.

All known Golgi glycosyltransferases are type II integral membrane proteins with three key features – a short amino-terminal cytosolic tail, a single transmembrane domain (TMD), and a lumenal region containing the catalytic domain *Welch and Munro, 2019*; *Banfield, 2011*; *Tu and Banfield, 2010*. Previous work from multiple groups have reported the individual contributions of cytosolic, transmembrane, and lumenal domains of glycosyltransferases in their localization *Burke et al., 1994*; *Colley,*

*1997*; *Fenteany and Colley, 2005*; *Banfield, 2011*; *Schmitz et al., 2008*; *Tu et al., 2008*; *Welch and Munro, 2019*; *Nilsson et al., 1993*; *Tu and Banfield, 2010*. However, how these domains contribute collectively to the steady-state localization of these Golgi enzymes is not clear.

Cisternal maturation is the most established model for Golgi organization and protein recycling, and proposes that Golgi resident proteins are specifically segregated away from anterograde and secretory cargos in the maturing cisternae and packaged into recycling vesicles *Pantazopoulou and Glick, 2019*; *Glick and Nakano, 2009*. Dynamic retrograde trafficking of the Golgi residents from maturing late Golgi to early cisternae is key to maintaining their localization *Banfield, 2011*; *Welch and Munro, 2019*. The recycling of several Golgi glycosylation and glycosphingolipid synthesis enzymes requires the activity of a phosphatidylinositol-4-phosphate [PI(4)P] binding protein, Vps74 (human GOLPH3), that binds to the COPI coat complex and to the cytosolic tails of its client Golgi enzymes *Schmitz et al., 2008*; *Tu et al., 2012*; *Tu et al., 2008*; *Rizzo et al., 2021*; *Welch et al., 2021*. The binding affinities between Vps74 and the cytosolic tails of glycosyltransferases are reported to be weak *Schmitz et al., 2008*, suggesting that successful retrieval in vivo may require more than binding to Vps74 alone. Considering the TMD and lumenal domains of glycosyltransferases have also been shown to affect recycling, an integral membrane protein capable of extended engagement with the glycosyltransferases, as well as of stabilizing the interactions with cytosolic recycling machinery can provide a key missing mechanistic link in this process.

*erd1* and *erd2* (ER Retention Defective) mutants were isolated three decades ago in a screen for mutants that secrete HDEL-invertase *Pelham et al., 1988*. Erd2 was subsequently characterized as the receptor that binds and retrieves ER proteins containing a C-terminal HDEL sequence, providing the underlying mechanism for its *erd* phenotype *Semenza et al., 1990*. Despite a strong defect in ER retention of HDEL-invertase and Kar2, as well as defects in Golgi-dependent modification of glyco-proteins observed in the *erd1* mutant, the function of Erd1 remains uncharacterized *Hardwick et al., 1990*; *Copic et al., 2009*. Here, we provide evidence that Erd1 chaperones early Golgi glycosylation enzymes throughout their trafficking in the Golgi, and facilitates stable interaction with the cyto-solic receptor Vps74, thus acting as a key component of the glycosyltransferase recycling pathway. Membrane-embedded Erd1 and cytosolic Vps74 are together required for successful recycling of a subset of enzymes from late Golgi to early Golgi compartments via COPI vesicles.

## Results and discussion
### Golgi localization of glycosyltransferases requires Erd1

We examined the genome-wide genetic interactions networks for several glycosyltransferases *Costanzo et al., 2016*, and observed significant profile similarity with Erd1. To uncover functional information from genetic network analysis, we performed spatial analysis of functional enrichment (SAFE) and generated a profile similarity network (PSN) for Erd1 *Usaj et al., 2017*; *Costanzo et al., 2016*; *Baryshnikova, 2016* (*Figure 1A and B*). SAFE analysis allows the identification of regions of the global similarity network that are significantly enriched for genes exhibiting negative or positive genetic interactions with a gene of interest. PSN analysis correlates the genetic interaction profiles and allows annotation of genes with related functions. Our analysis indicated that Erd1 is functionally related to genes known to mediate glycosylation and transport events at the ER and Golgi.

*erd1Δ* mutant cells exhibited sensitivity to the aminoglycoside hygromycin, consistent with defects in glycosylation *Ballou et al., 1991*; *Dean, 1995* (*Figure 1C*, *Figure 1—figure supplement 1A*), as well as tunicamycin, an inhibitor of N-linked glycosylation (*Figure 1D*). Additionally, as compared to WT control cells, we observed altered mobility of reporters of O-linked (Gas1) and N-linked glycosylation (CPY/Prc1, Pep4) in *erd1Δ* lysates, indicative of defects in glycosylation at the Golgi (*Figure 1—figure supplement 1B*). These observations are also in agreement with the reported defects in Golgi based CPY and invertase modifications in the *erd1* mutant isolated from the HDEL-invertase screen *Hardwick and Pelham, 1990*. Endogenously tagged Erd1-mNeongreen predominantly co-localized with the early-Golgi marker, Mnn9, and to some extent with the medial-Golgi marker, Gea2, suggesting its function in early to medial Golgi compartments (*Figure 1E*, *Figure 1—figure supplement 1C*).

To assess the role of Erd1 in protein trafficking, we examined the localization of several represen-tative ER, cis-, medial-, and trans-Golgi proteins tagged with a green fluorescent protein (GFP) in WT and *erd1Δ* cells. All the proteins tested exhibited the expected ER or punctate localization in WT

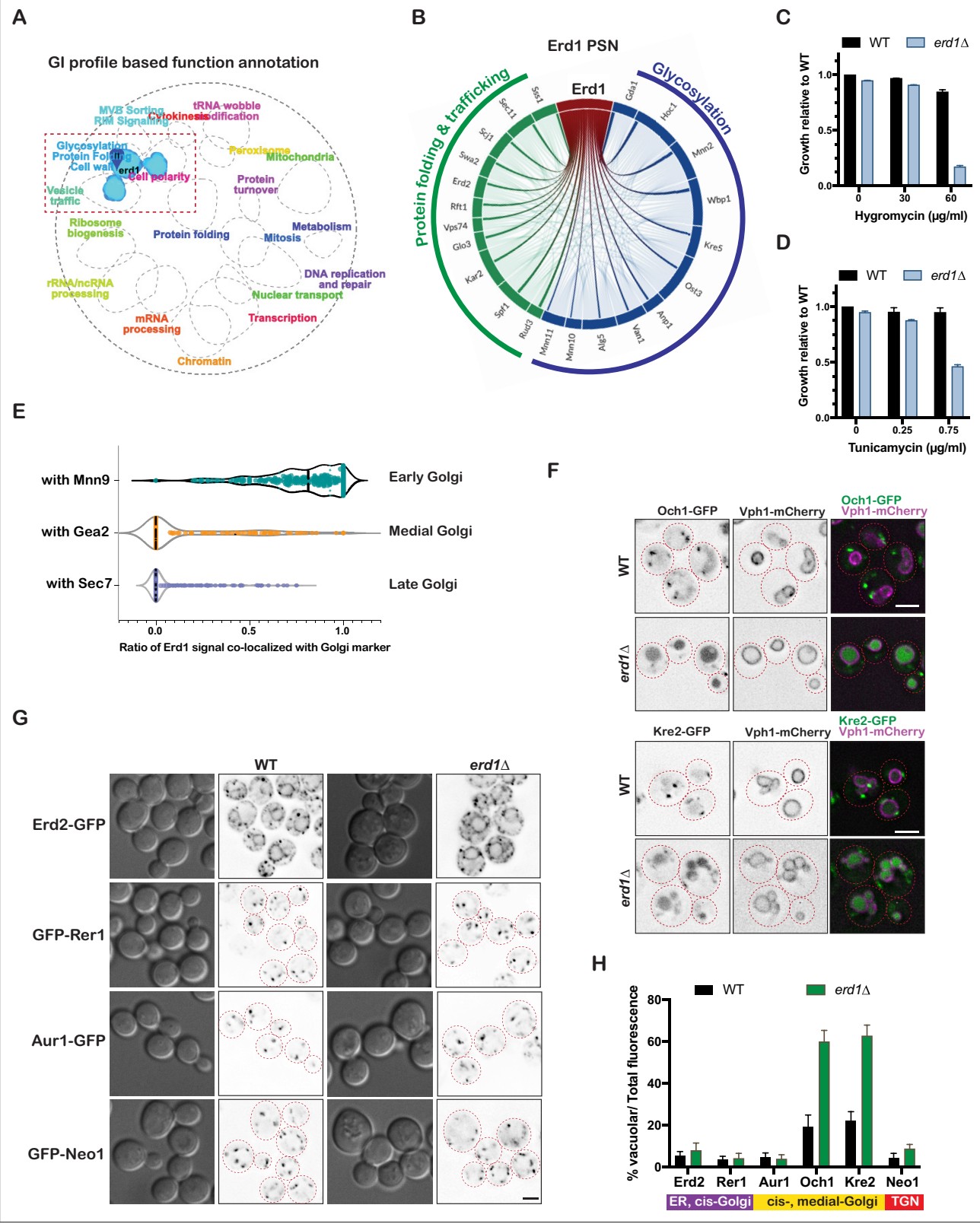

**Figure 1.** Erd1 is required for Golgi protein glycosylation. (**A**) Spatial analysis of functional enrichment (SAFE) analysis based on the genetic interaction profile of *erd1Δ* mutant (stringent cut-off (p < 6e-11)). (**B**) Profile similarity network (PSN) of Erd1 showing genes with similar genetic interactions (similarity cut-off 0.25). (**C**) Growth of wild type and *erd1Δ* mutant in the presence of indicated concentrations of HygromycinB in YPD liquid cultures at 30° C after 24 hr. (**D**) Sensitivity of wild type and *erd1Δ* mutant to indicated concentrations of Tunicamycin in YPD liquid cultures at 30° C. (**E**) Violin plots

*Figure 1 continued on next page*

Figure 1 continued

for the ratio of co-localized Erd1-mNeonGreen fluorescence with early (Mnn9-mCherry), medial (Gea2-3xmMars), and late (Sec7-6xDsRed) Golgi markers (n = 250 puncta for each condition). The median is indicated with dashed lines. (**F**) Live-cell fluorescence imaging of Och1-GFP, Kre2-GFP, and vacuole membrane marker (Vph1-mCherry) in wild type and *erd1Δ* mutant. Red dashed lines indicate the cell boundaries based on DIC images. (**G**) Live-cell fluorescence imaging of Erd2-GFP, GFP-Rer1, Aur1-GFP and GFP-Neo1 in wild type and *erd1Δ* mutant. (**H**) Quantification of the percent vacuolar to total GFP fluorescence for reporters in (**F**) and (**G**). Scale bars: 2.5µ m.

The online version of this article includes the following figure supplement(s) for figure 1:

**Figure supplement 1.** Erd1 localizes to the cis-Golgi and is required for protein glycosylation.

cells; however, the Golgi glycosyltransferases, Och1 and Kre2, were significantly mislocalized to the vacuole lumen in the *erd1Δ* mutant (*Figure 1F, G and H*). A similar mislocalization of Och1, but not Mnn9, was previously observed in the *erd1Δ* mutant *Okamoto et al., 2008*. This suggested a specific sorting defect in *erd1Δ* mutant cells rather than a more general Golgi trafficking defect.

## Erd1 is required for Vps74 function in intra-Golgi recycling

The defect in glycosyltransferase recycling is highly reminiscent of a similar defect previously reported for the *vps74Δ* mutant *Schmitz et al., 2008*; *Tu et al., 2008*. Vps74 is a peripheral membrane protein that acts as the cytosolic receptor for COPI recycling *Schmitz et al., 2008*; *Tu et al., 2012*; *Tu et al., 2008*; *Welch et al., 2021*; *Rizzo et al., 2021*. To assess the relationship between Vps74 and Erd1, we first directly compared the glycosylation defects in *erd1Δ* and *vps74Δ* mutants. The *erd1Δ* and *vps74Δ* mutants phenocopied each other in the Gas1 glycosylation defect, as well as in tunicamycin and hygromycin sensitivities (*Figure 2A, B*, *Figure 2—figure supplement 1A*). Furthermore, SAFE analysis on the global genetic interactions indicated a robust functional overlap between Erd1 and Vps74 (*Figure 2—figure supplement 1B*). Simultaneous deletion of Erd1 and Vps74 did not worsen the temperature sensitive slow growth of *vps74Δ* mutant at 40° C suggesting the two act in a common pathway (*Figure 2C*, *Figure 2—figure supplement 1A*). While the specific Gas1 glycosylation defect in *erd1Δ* and *vps74Δ* mutants was most similar to the loss of Kre2 (*Figure 2—figure supplement 1C*), their sensitivity to hygromycin resembled a defect similar to the loss of multiple glycosyltransferases (*Figure 2—figure supplement 1D*). To test this directly, we examined the localization of a panel of endogenously tagged glycosyltransferases and early-Golgi proteins in WT, *erd1Δ*, and *vps74Δ* mutants (*Figure 2D*). Indeed, loss of Erd1 and Vps74 manifested in a defect in recycling of the same subset of Golgi glycosyltransferases. All the tested glycosyltransferases that were Erd1-dependent also contained the cytosolic motif previously reported to require Vps74 binding (*Figure 2—figure supplement 2E*) *Tu et al., 2008*.

To investigate the mechanism of interdependence of Erd1 and Vps74 function, we first asked if they were required for each other's correct localization. In wild-type cells, Erd1-GFP exhibited punctate distribution similar to cis-Golgi markers (*Figure 1—figure supplement 1C*). Strikingly, Erd1-GFP was missorted to the vacuole lumen in the *vps74Δ* mutant, as indicated by the co-localization with the vacuole lumen stained with the fluorescent dye, CMAC (*Figure 2E*). This suggests that in the absence of Vps74, Erd1 itself is not incorporated into COPI recycling vesicles and is instead delivered to the vacuole. On the other hand, endogenously tagged mNG-Vps74 co-localized predominantly with the medial-Golgi marker, Gea2, in both WT and *erd1Δ* mutant (*Figure 2F*, *Figure 2—figure supplement 1F and 1G*). Thus, whereas Erd1 is not required for recruitment of Vps74, it appears to be required for the function of Vps74 at the Golgi. Even though Vps74 is recruited to the Golgi via its interaction with PI4P *Wood et al., 2009*, it is unable to mediate recycling in the *erd1ΔΔ* mutant. These observations also indicate that the Vps74-dependent pathway recycles single pass type II membrane proteins such as glycosyltransferases as well as multi-pass transmembrane proteins such as Erd1.

If Erd1 and Vps74 cooperate to facilitate recycling, we asked if overexpression of Erd1 and Vps74 can compensate for each other in functional assays. Overexpression of Erd1 in the *vps74Δ* mutant partially suppressed the hygromycin and temperature sensitivity of the *vps74Δ* mutant (*Figure 2G*). In contrast, overexpression of Vps74 in the *erd1Δ* mutant, severely inhibited its growth fitness (*Figure 2I*). This toxicity was specifically due to maintenance of Vps74 expressing plasmid, as the growth was normal on rich media with no selection for the plasmid. Mutations in Vps74 that disrupt COPI binding (*vps74 R6-8A*) or oligomerization (*vps74Δ201–204*) alleviated the toxicity (*Figure 2H*, *Figure 2—figure supplement 1E*). One interpretation of these results is that overexpression of Vps74

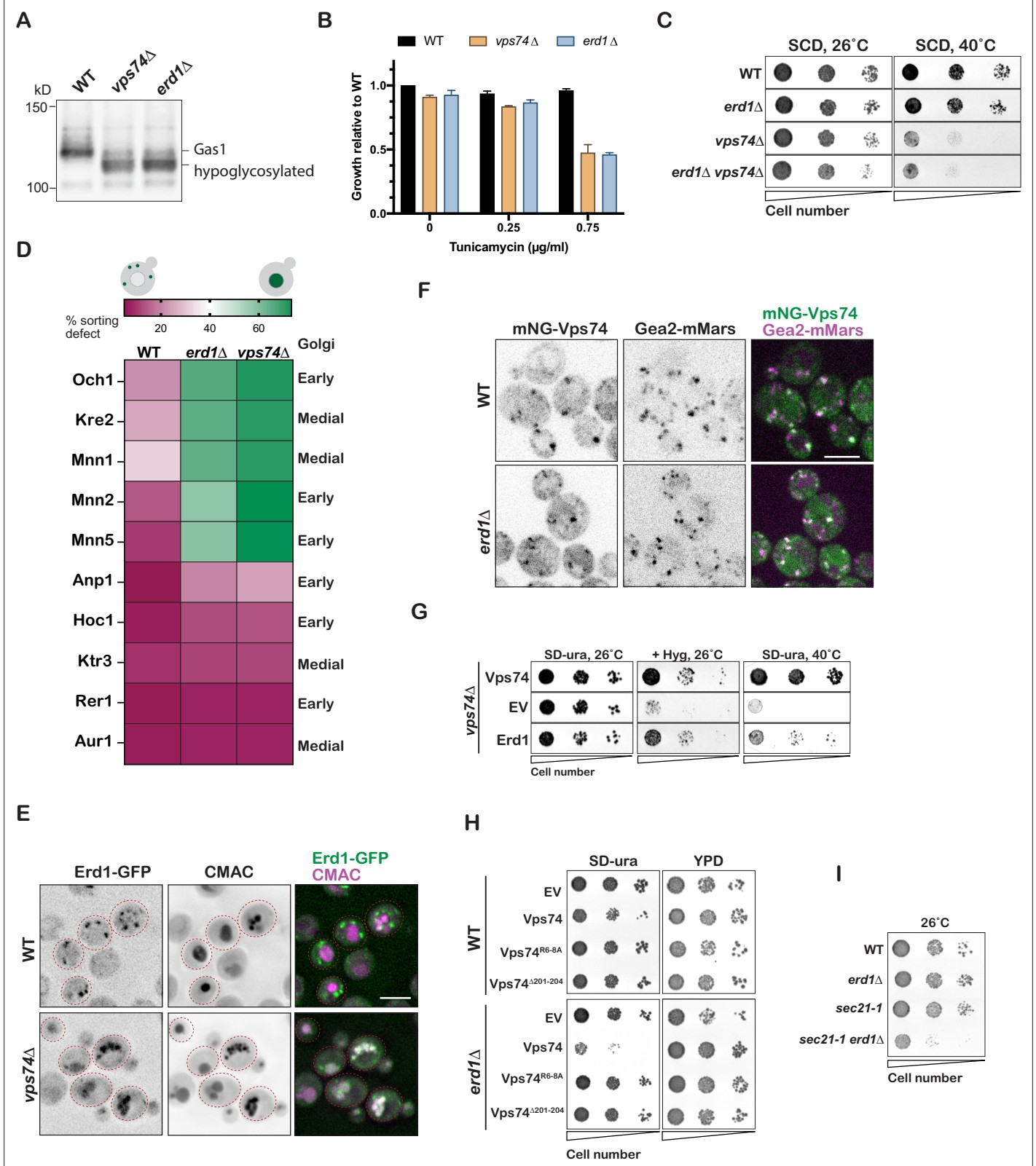

**Figure 2.** Erd1 is required for Vps74-COPI dependent recycling of specific Golgi glycosyltransferases. (**A**) Immunoblot analysis on yeast cell lysates from wild type, *erd1Δ*, and *vps74Δ* mutant for glycosylation reporter, Gas1. (**B**) Growth of wild type, *erd1Δ*, and *vps74Δ* mutant in the presence of indicated concentrations of Tunicamycin in YPD liquid cultures at 30° C after 24 hr. (**C**) Growth of serial dilutions of wild type, *erd1Δ*, *vps74Δ*, and *erd1Δvps74Δ* mutants on synthetic media at 26 °C and 40° C after 2 days. (**D**) Quantification of percent vacuolar fluorescence to total fluorescence of the indicated

*Figure 2 continued on next page*

*Figure 2 continued*

GFP tagged early and medial Golgi proteins in wild type, *erd1Δ*, and *vps74Δ* mutant. (**E**) Live-cell fluorescence imaging of Erd1-GFP and vacuolar dye, CMAC in wild type and the *vps74Δ* mutant. (**F**) Live-cell fluorescence imaging of mNeonGreen-Vps74 and medial Golgi marker, Gea2-3xMars in wild type and the *erd1Δ* mutant. (**G**) Growth of serial dilutions of *vps74Δ* mutant transformed with empty vector (EV) or plasmids overexpressing Vps74 and Erd1 on YPD with 50 μg/ml hygromycinB at 26 °C or synthetic media lacking uracil at 26 °C or 40° C after 2–3 days. (**H**) Growth of serial dilutions of wild type and *erd1Δ* mutant transformed with the indicated Vps74 mutants at 26° C after 3 days. (**I**) Growth of serial dilutions of wild type, *erd1Δ*, *sec21-1*, and *sec21-1 erd1Δ* mutants at 26 °C after 3 days. Scale bars: 2.5μ m.

The online version of this article includes the following figure supplement(s) for figure 2:

**Figure supplement 1.** Erd1 and Vps74 mutants exhibit similar glycosylation defects.

**Figure supplement 2.** Pmr1 is required for glycosylation but not glycosyltransferase recycling.

in the *erd1Δ* mutant resulted in futile Vps74-COPI complexes that reduced the level of free COPI available for Golgi function. Consistent with this, the *erd1Δsec21-1* mutant, defective in Erd1 and COPI function, exhibited a similar synthetic growth defect even at permissive temperature (*Figure 2I*). Collectively, these observations indicate significant interdependence between Erd1 and Vps74 in the COPI-mediated intra-Golgi recycling of specific Golgi cargos.

Erd1 has previously been proposed to play a role in Golgi ion homeostasis by transporting phosphate from the Golgi lumen to the cytosol *Snyder et al., 2017*. The *erd1Δ* mutant led to loss of phosphate (Pi) via exocytosis, and required 1.5-fold higher concentration of Pi in the media for 50% maximal growth as compared to wild type *Snyder et al., 2017*. To test if the effect of Erd1 on glycosyltransferase recycling is distinct from effects on ionic balance, we compared the *erd1Δ* phenotypes with that of *pmr1Δ* mutant. Pmr1 pumps Ca2+ and Mn2+ ions into the Golgi lumen *Dürr et al., 1998*. As expected, deletion of Pmr1 results in a similar Gas1 glycosylation defect as loss of Erd1 or Vps74 (*Figure 2—figure supplement 2A*). However, while supplementing the media with Ca2+ or Mn2+ rescued the glycosylation defect observed in *pmr1Δ* mutant (*Figure 2—figure supplement 2C*), supplementing the media with even 50-fold higher Pi did not rescue the glycosylation defect in the *erd1Δ* mutant. Importantly, despite the glycosylation defect, *pmr1Δ* mutant had no defect in the localization and recycling of Kre2 (*Figure 2—figure supplement 2D*). Therefore, while it is possible that Erd1 plays a role in phosphate homeostasis at the early Golgi, our data support a distinct role of Erd1 in also being required for glycosyltransferase recycling.

## Erd1 directly interacts with glycosyltransferases and Vps74 to facilitate recycling

Based on our observations, we predicted that Erd1 directly interacts with Vps74 as well as glycosyltransferases. To monitor interactions between Erd1 and Vps74, we employed bimolecular fluorescence complementation (BiFC), co-immunoprecipitation, and split-ubiquitin based yeast two-hybrid (Y2H) assays *Iyer et al., 2005*; *Feng et al., 2017*. We made fusions to the N-terminal beta strands 1–10 of mNeongreen, and the C-terminal beta strand 11 of mNeongreen, and since Vps74 has been shown to oligomerize, it served as a positive control in the BiFC assay (*Figure 3A*) *Cai et al., 2014*; *Wood et al., 2012*; *Schmitz et al., 2008*. We observed fluorescence complementation between Erd1-NG11, and NG1-10-Vps74 (*Figure 3A*). Fluorescence complementation was also detectable between tagged Erd1 and COPI binding defective *vps74 R6-8A* mutant, but not between tagged Erd1 and the oligomerization defective *vps74 Δ201–204* mutant. These data suggest that Erd1 and Vps74 interaction is likely not bridged via COPI binding. Erd1 and Vps74 interaction was also confirmed using co-immunoprecipitation and Y2H assays (*Figure 3B*, *Figure 3—figure supplement 1A*). Erd1 also showed interaction with itself in Y2H and co-immunoprecipitation assays, suggesting Erd1 likely oligomerizes in vivo (*Figure 3—figure supplement 1A and B*).

To test interaction with glycosyltransferases, we immunoprecipitated GFP tagged Erd1 and Vps74 from yeast cell lysates and probed for Och1-FLAG in the pulldown by immunoblotting (*Figure 3C*). Och1 co-immunoprecipitated with both Erd1 and Vps74. Since tagging the cytosolic N-terminus of the glycosyltransferases resulted in retention in the ER (*Figure 3—figure supplement 1D*), we tagged the lumenal facing ends of Och1 and Erd1 to test interaction using BiFC. We observed robust fluorescence complementation signal between Och1-NG1-10 and NG11-Erd1, that co-localized with the cis-Golgi marker Mnn9 (*Figure 3A*). We next asked if the interaction between Erd1 and glycosyltransferases

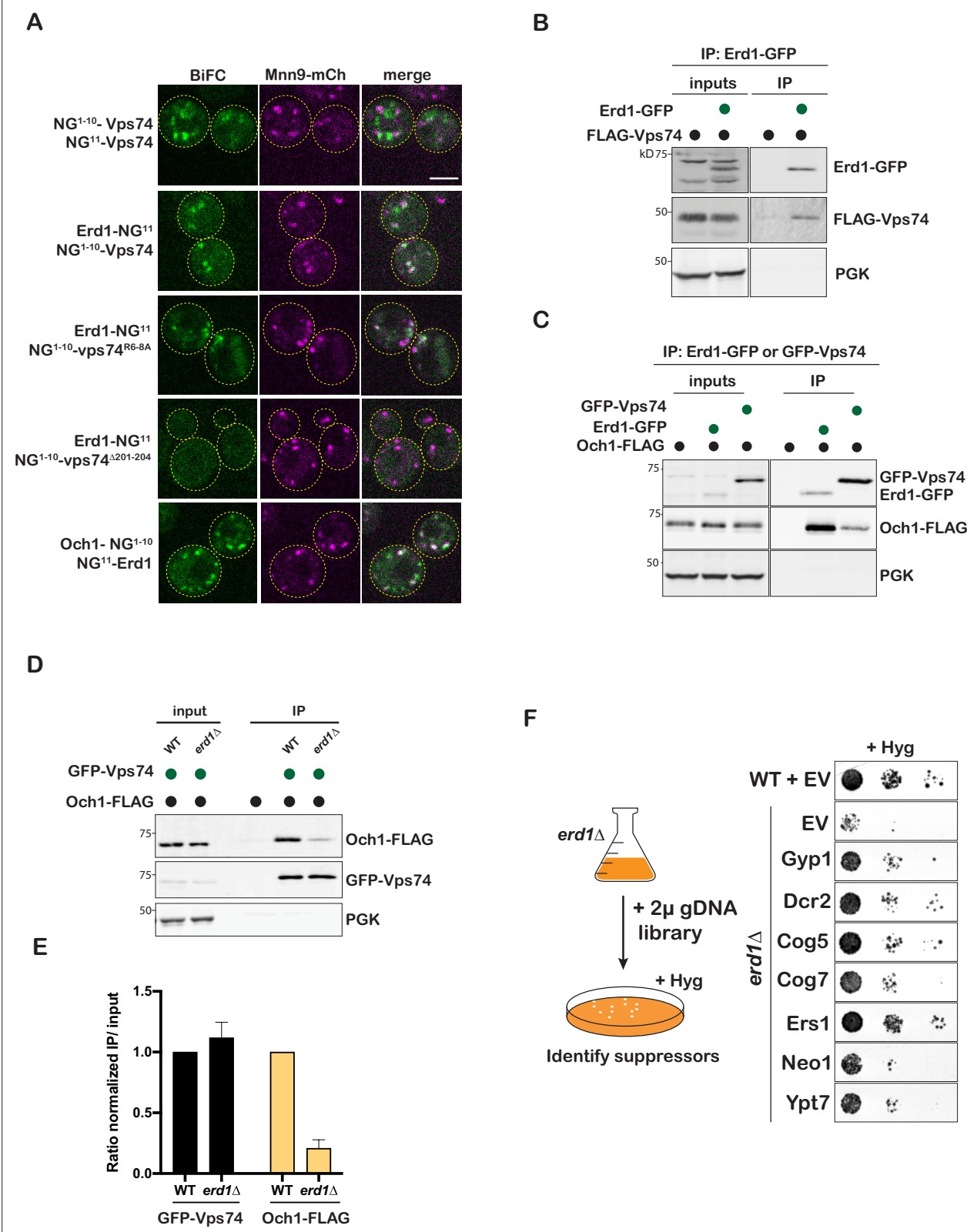

**Figure 3.** Erd1 interacts with glycosyltransferases and Vps74. (**A**) Bimolecular fluorescence complementation of mNeongreen tested in cells with plasmids expressing split-mNG fragments fusions to the indicated proteins, in a strain also expressing the early Golgi marker, Mnn9-mCherry. BiFC signal is shown in green, and Mnn9-mCherry is shown in magenta. (**B**) Coimmunporecipitation analysis to test the interaction between Erd1-GFP and FLAG-Vps74 (FLAG tag listed in all cases is 6 x His-TEV cleavage site-3xFLAG) from yeast cell lysates. (**C**) Coimmunporecipitation analysis to test the

*Figure 3 continued on next page*

*Figure 3 continued*

interaction between Erd1-GFP and Och1-FLAG, and GFP-Vps74 and Och1-FLAG from yeast cell lysates. (**D**) Coimmunoprecipitation analysis to test the interaction between GFP-Vps74 and Och1-FLAG from wild type and *erd1Δ* yeast cell lysates. (**E**) Quantification of ratio of the signal in the IP to input normalized to the loading control for GFP-Vps74 and Och1-FLAG. (**F**) Growth of serial dilutions of *erd1Δ* mutant transformed with plasmids expressing Gyp1, Dcr2, Cog5, Cog7, Ers1, Neo1, Ypt7 identified in the dosage suppressor screen on YPD with 60 µg/ml hygromycin at 26 °C for 2 days.

The online version of this article includes the following figure supplement(s) for figure 3:

**Figure supplement 1.** Erd1 is required for Golgi protein trafficking.

may be required for stable complex formation with Vps74. Although GFP-Vps74 efficiently coimmunoprecipitated Och1-FLAG or Kre2-FLAG from WT yeast cell lysates, the interaction was dramatically reduced in the *erd1Δ* mutant (*Figure 3D,E*, *Figure 3—figure supplement 1E*). Overall, these findings support a model where Erd1 chaperones the Golgi enzymes and allows stable interaction with the cytosolic recycling machinery.

To identify mechanisms that can compensate for the loss of Erd1 function, we employed a dosage suppressor screen to identify genes which when expressed at a higher than normal level can substitute for Erd1 function or bypass its requirement. We transformed the *erd1Δ* mutant with a multicopy yeast gDNA library *Herman et al., 1991*, isolated and confirmed suppressors of its hygromycin sensitivity (*Figure 3F*). Interestingly, we identified several genes with known roles in membrane protein trafficking (Gyp1, Cog5, Cog7, Ypt7) as well as those with previously reported genetic interactions with either Erd1 or Vps74 (Dcr2, Neo1, Ers1) *Wang et al., 2020*; *Miyasaka et al., 2020*; *Hardwick and Pelham, 1990*. The suppression was specific to *erd1Δ* mutant, since the introduction of these plasmids did not suppress the growth inhibition of wild type cells at high concentrations of hygromycin (*Figure 3—figure supplement 1C*). Gyp1 is a GTPase-activating protein for the Rab GTPase Ypt1 at the cis-Golgi, and it negatively regulates Ypt1 *Du and Novick, 2001*. Consistent with this, we found that the GDP-locked (inactive) mutant of Ypt1 (*ypt1 S22N*) but not WT Ypt1 or the GTP-locked (active) form (*ypt1 Q67L*) partially suppressed the hygromycin sensitivity of the *erd1Δ* mutant, suggesting that reduction in Ypt1 function is the basis of Gyp1 suppression (*Figure 3—figure supplement 1F*). Also intriguing is the suppression by subunits of the COG complex which is known to be required for the tethering of vesicles for intra-Golgi glycosyltransferase recycling *Pokrovskaya et al., 2011*; *Ungar et al., 2006*. Overall, the hits identified from the multicopy suppressor analysis support a role of Erd1 in Golgi protein recycling.

## Erd1 and Vps74 sequentially engage with the glycosyltransferases

At steady state Erd1 and the glycosyltransferases localize predominantly at the early Golgi, while Vps74 localizes to the medial and late Golgi compartments. If Erd1 and Vps74 function cooperatively in mediating glycosyltransferase recycling, they would be expected to exhibit temporal overlap during Golgi maturation. To investigate this, we examined the in vivo spatiotemporal dynamics of Erd1 and Vps74 relative to each other, and other markers of Golgi cisternal maturation. Endogenously tagged constructs of Vps74, Erd1, and Och1 were confirmed to be functional (*Figure 4—figure supplement 1D*). Using time-lapse imaging to track individual Golgi puncta, we found that Erd1 arrived and departed in synchrony with the Golgi glycosyltransferases Mnn9 and Och1 (*Figure 4A,B*, *Figure 4—figure supplement 1A,B*). Erd1 signal peaked at the same time as Mnn9, and significantly before the medial (Gea2) and late Golgi (Sec7) markers (*Figure 4C,D*, *Figure 4—figure supplement 1E*). In contrast, Vps74 levels peaked after Gea2, and before Sec7 or the PI4P-specific probe, P4C *Luo et al., 2015*; *Highland et al., 2021* (*Figure 4E*, *Figure 4—figure supplement 1C,F*). Interestingly, COPI signal peaked after Erd1, but before the arrival of Vps74 (*Figure 4F*). In agreement with these observations, simultaneous kinetic analysis of Erd1 and Vps74 also indicated temporal overlap between the two (*Figure 4G*).

The time-lapse imaging revealed that Erd1 has the same kinetics as its recycling clients, and the subsequent arrival of Vps74 coincides with recycling. These observations are consistent with sequential association of Erd1 and Vps74 with their client cargos and packaging into COPI vesicles (*Figure 4H*). Our data support a model where engagement of Erd1 with the membrane bound cargos allows stable interaction with Vps74, and commits them for recycling (*Figure 4I*). By chaperoning the early Golgi enzymes throughout the recycling process, Erd1 allows productive coordination of Vps74 binding with the timing of cargo recycling in the spatio-temporal context of Golgi cisternal maturation. Erd1 itself is

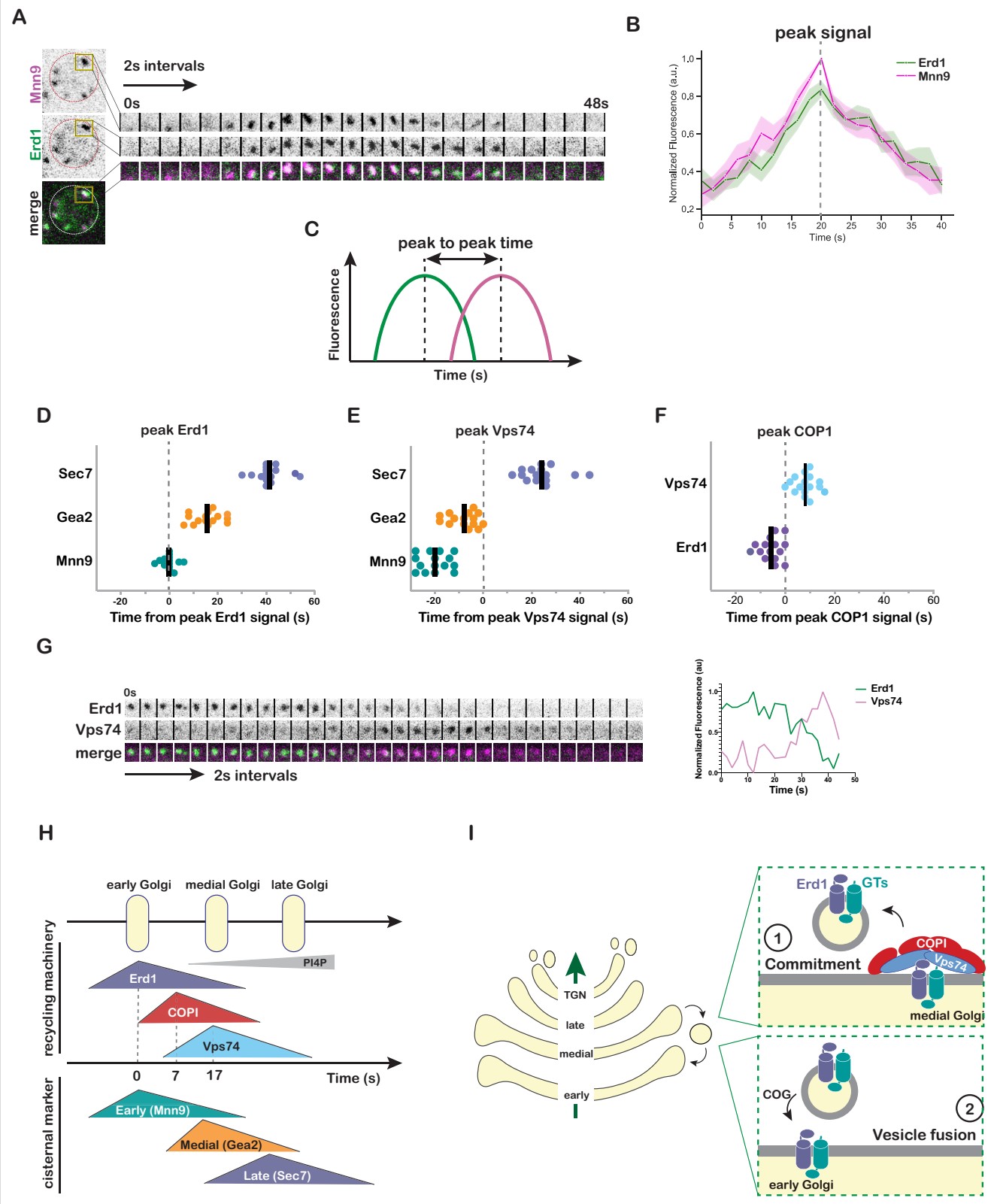

**Figure 4.** Kinetic analysis of Erd1 and Vps74 in the context of Golgi cisternal maturation. (**A**) Time-lapse imaging series (2 s intervals) of the indicated single Golgi compartment in cells expressing Erd1-mNeonGreen and Mnn9-mCherry. (**B**) Averaged normalized fluorescence traces with 95% CI of time-lapse imaging analysis of 10 puncta described in (**A**). Dashed line represents the time corresponding to the peak signal of Erd1 and Mnn9. (**C**) Schematic describing the estimation of peak-to-peak fluorescence times of two markers from time-lapse imaging analysis. (**D**) Quantification of peak-to-peak times

*Figure 4 continued on next page*

*Figure 4 continued*

for Erd1-mNeonGreen with respect to early (Mnn9-mCherry), medial (Gea2-3xMars), and late (Sec7-6xDsRed) Golgi markers (n > 15 puncta for each condition). Peak Erd1 is set at t0 and indicated with the dashed line. The bold line represents the median value for each condition. (**E**) Quantification of peak-to-peak times for mNeonGreen-Vps74 with respect to early (Mnn9-mCherry), medial (Gea2-3xMars), and late (Sec7-6xDsRed) Golgi markers. Peak Vps74 is set at t0 and indicated with the dashed line. The bold line represents the median value for each condition. (**F**) Quantification of peak-to-peak times for Erd1 and Vps74 with respect to COP1-mCherry. Peak COP1 is set at t0 and indicated with the dashed line. (**G**) Time-lapse imaging series (2 s intervals) of the indicated single Golgi compartment in cells expressing Erd1-mNeonGreen and mScarlet-Vps74 (left) and normalized fluorescence of time-lapse imaging (right). (**H**) Schematic depicting the dynamics of tagged Erd1, Vps74 and COP1 at maturing Golgi compartments marked by Mnn9 (early), Gea2 (medial), and Sec7 (late) based on A-F. (**I**) A simplified model for the proposed role of Erd1 in the recycling of Golgi glycosyltransferase at two steps - (1) at the step of commitment of the cargos for recycling by allowing stable complex formation with the cytosolic recycling machinery, and (2) at the final step of vesicle tethering and fusion at the early Golgi by facilitating interaction with the tethering machinery.

The online version of this article includes the following figure supplement(s) for figure 4:

**Figure supplement 1.** Kinetic analysis of Erd1 and Vps74 with Golgi markers.

also a cargo of the Vps74 pathway, and it is possible that Erd1's role in Golgi recycling could be more complex and it may be involved in distinct steps for Golgi to ER Vps74-independent and intra-Golgi Vps74-dependent recycling steps.

Overall, our work demonstrates a role of Erd1 in Golgi protein recycling, and provides a mechanistic basis for the observed glycosylation defects. This work also provides new insight on how cooperative, yet hierarchical engagement of transmembrane (Erd1) and cytosolic receptors (Vps74), likely with different domains of their transmembrane cargos, is important for productive recycling. By extended engagement with the cargos, and with the cytosolic recycling machinery, Erd1 might serve as an important quality assurance sensor at multiple steps in the pathway. Other such transmembrane adaptors are also likely to exist. Recent screens in mammalian cells have identified transmembrane proteins TMEM165 and TM9SF2 that may serve similar roles in bridging the Golgi Gb3 synthase A4GALT to cytosolic recycling machinery *Yamaji et al., 2019*; *Tian et al., 2018*; *Tanaka et al., 2017*. Similarly, transmembrane Erv proteins can serve as receptors for COPII dependent ER to Golgi protein trafficking *Barlowe and Miller, 2013*. A lot remains to be discovered to understand the role and the interplay of different components at multiple steps of Golgi enzyme recycling- vesicle tethering complexes such as the COG and GARP complexes, as well as Golgi matrix proteins such as GRASP55 are also required for recycling Golgi glycosylation enzymes *Khakurel et al., 2021*; *Pokrovskaya et al., 2011*; *Pothukuchi et al., 2021*. Suppression of the *erd1Δ* mutant by COG subunits also suggests a potential role for Erd1 at the final tethering step of the recycling process that would need further examination.

The role of Erd1 and other EXS domain proteins such as human XPR1 and plant PHO1 in phosphate homeostasis has been previously reported, although unlike Erd1, XPR1 and PHO1 both also contain a phosphate sensing SPX domain *Secco et al., 2012*; *Wege et al., 2016*; *Giovannini et al., 2013*; *Snyder et al., 2017*; *Legati et al., 2015*; *Bondeson et al., 2020*; *Arpat et al., 2012*. XPR1 functions in phosphate transport at the cell surface, but also localizes intracellularly *Bondeson et al., 2020*. PHO1 localizes to the Golgi and trans-Golgi network *Arpat et al., 2012*, and it is possible that the intracellular pools of PHO1 and XPR1 also play roles in protein trafficking. Since glycosylation reactions in the Golgi produce inorganic phosphate (Pi) and protons (H+) as byproducts, these proteins could also help maintain glycosylation homeostasis more broadly. Future analysis will examine how Erd1 engages with only a subset of client glycosyltransferases and with Vps74. A systematic comparison of Erd1 and Vps74 cargo repertoire will also be important to determine the extent of overlap between their clients. Finally, it remains to be determined whether transmembrane proteins in recycling vesicles, such as Erd1, play a role in decisions that allow some COPI retrograde vesicles to be recycled within the Golgi, while others are trafficked to the ER. These and other exciting questions remain to be addressed as we move forward.

## Materials and methods
### Yeast strains and plasmids
All yeast deletion or chromosomal tagged strains were constructed in the strain background SEY6210 using iterative markerless CRISPR/Cas9 genome engineering, or using PCR-based amplification and homologous recombination as described *Shaw et al., 2019*; *Longtine et al., 1998*. All yeast strains

used in this study are listed in *Supplementary file 1*. Cells were grown in yeast synthetic media [0.17 % (w/v) yeast nitrogen base, 0.5 % (w/v) ammonium sulfate, 2 % (w/v) glucose, and supplemented with appropriate nutrients] or YPD [1 % (w/v) yeast extract, 1 % (w/v) peptone, 2 % (w/v) glucose] at 26 °C, unless otherwise specified. All plasmids used in this study are listed in *Supplementary file 1*, and were generated using standard cloning procedures. Yeast transformations were performed following the standard lithium acetate protocol *Schiestl et al., 1993*.

## Growth assays

For growth assays, ten-fold serial dilutions of mid-log phase cultures normalized to OD 0.3 were spotted onto the indicated media and incubated for 2–5 days. Y2H assay was performed in yeast strain L40 by co-transforming the indicated bait and prey vectors. Interaction was scored as growth on synthetic media lacking Leu, Trp and His and containing 1 mM 3-Amino-1,2,4-triazole (3-AT). Dosage suppressor screen was performed by transforming a 2 micron URA3 gDNA libray into *erd1Δ* mutant and suppressors were selected on YPD plates containing 75 µg/ml HygromycinB. The suppressing plasmids were extracted, amplified in bacteria, re-transformed into *erd1Δ* yeast cells and the suppressing genes were identified by sequencing.

## Genetic interaction analysis

Analysis of genetic interactions was performed on the genome-wide genetic interaction dataset available on the CellMap repository *Usaj et al., 2017*. SAFE analysis was performed as described *Baryshnikova, 2016* using negative genetic interactions and stringent cut-off settings (p < 6e-11). Erd1 PSN was generated using a threshold cutoff off 0.25 and visualized as a Chord plot.

## Fluorescence microscopy

Cells were grown in synthetic dropout media to mid-log phase at 26° C and imaged at room temperature on glass coverslips or in glass-bottomed dishes. Images unless mentioned otherwise were captured using a DeltaVision RT system (Applied Precision, Issaquah, WA), equipped with a DV Elite CMOS camera, a 100 x objective, and a DV Light SSI 7 Color illumination system with Live Cell Speed Option (FITC for GFP and TRITC for mCherry/RFP). Image acquisition and deconvolution were performed using the provided DeltaVision software (softWoRx 6.5.2; Applied Precision, Issaquah, WA). Masks for vacuoles and the total cell were created based on DIC images in ImageJ. Mean vacuolar to total fluorescence ratios were calculated from three independent experiments with ~25 cells in each set.

All time-lapse images were captured with a CSU-X spinning-disk confocal microscope system (Intelligent Imaging Innovations) using a DMI6000 microscope (Leica Microsystems) outfitted with a CSU-X1 spinning-disk confocal unit (Yokogawa Electric Corporation) with a QuantEM 512SC (Photometrics). The objective was a 100 × 1.46 NA Plan Apochromat oil immersion lens (Leica Microsystems). Images were captured every 2 s for 2 min using Slidebook six software (Intelligent Imaging Innovations). Images were processed in ImageJ and assembled in Adobe Illustrator CS6. Quantification of Golgi puncta was perfomed in Python. Briefly, images were corrected for hot pixels by using a median filter, and uneven illumination by applying a gaussian background subtraction. Puncta in both channels were defined by thresholding using the Otsu method. Object based co-localization was used to calculate the ratio of overlap between the two channels for ~250 puncta for each set. The quantification data of each puncta was plotted, and the median and confidence intervals were annotated in violin plots. For generation of time lapse imaging traces, individual Golgi compartments that remained in the same focal plane throughout their lifetimes were used for analysis. Fluorescence intensities for the identified puncta were measured over time and intensity values were normalized to values between 0 and 1 for each channel. Peak-to-peak time values denote the time between maximum intensities of green and red fluorescence signals at individual maturing Golgi puncta. Averaged traces with confidence intervals were generated from ~10 puncta for each channel.

## Protein extraction and immunoprecipitation

Yeast lysates for immunoblotting were prepared from 5 OD600 equivalents of cells. Briefly, cells were harvested and incubated in 10 % trichloroacetic acid (TCA) on ice for 1 hr. Cells were pelleted, washed with cold acetone, lysed by beating with glass beads in 80 µl of 2 x urea buffer (50 mM Tris-HCl pH 7.5, 8 M urea, 2 % SDS, 1 mM ethylenediaminetetraacetic acid [EDTA]) for 5 min at room temperature,

followed by incubation at 42 °C for 10 min. 80 µl of 2 x sample buffer (150 mM Tris HCl, pH 6.8, 8 M urea, 8 % SDS, 24 % glycerol, bromophenol blue) supplemented with 100 mM dithiothreitol (DTT) was added, and samples were vortexed for 5 min followed by incubation at 42 °C for 10 min. The samples were centrifuged and 0.2 OD equivalents were resolved on eight or 10% SDS-Polyacrylamide gels, followed by transfer to nitrocellulose membranes (0.45 µm, GE healthcare) at 4 °C via wet transfer in transfer buffer (25 mM Tris, 192 mM Glycine, 10% v/v methanol, 0.006 % SDS) at 100 V for 90 min before immunoblotting. For glycosylation rescue experiments, cells were grown to mid-log phase in YPD at 30 °C, followed by growth in YPD supplemented with 50 mM CaCl2, 600 µM MnCl2, or 50 mM phosphate buffer pH 6.2 for 4 hr at 30 °C.

For immunoprecipitations, cells were grown to OD600 0.5–1.0 in 100 ml of synthetic growth media and harvested on ice. Cells were washed and resuspended in Lysis Buffer (50 mM Tris pH 7.5, 2 mM EDTA pH 8.0, 150 mM NaCl, 10 % glycerol, 1 mM PMSF, 1 X Roche cOmplete Protease Inhibitor Tablet/50 ml). Cell extracts were prepared by glass bead beating for three cycles of 1 min vortexing with 1 min breaks on ice. Membranes were solubilized by nutating for 30 min at 4 °C after addition of Saponin to 1 % final concentration. Crude extracts were clarified and the lysates was incubated with 25 µl of GFP-nanobody resin at 4 °C for 3 hr. The resin was washed five times with Lysis buffer and bound proteins were eluted by addition of 100 µl 2 x sample buffer (100 mM Tris-HCl pH 8.0, 1 % SDS, 10 mM DTT), followed by incubation at 65 °C for 10 min. 1% input and 10% immunoprecipitates were resolved on a 10% SDS-PAGE gel and subjected to immunoblotting.

The following antibodies and dilutions were used for western blotting: rabbit polyclonal anti-Gas1 (1:20000), rabbit polyclonal anti-CPY (1:5000), rabbit polyclonal anti-Pep4 (1:2000), rabbit polyclonal anti-GFP (1:5000) (TP401; Torrey Pines Biolabs, Secaucus, NJ), mouse monoclonal anti-PGK (1:5000) (22C5D8; Molecular Probes Inc), mouse monoclonal anti-FLAG (1:3000) (M2; Sigma), 800CW goat anti-rabbit (1:10,000) (926–32211; LI-COR Biosciences, Lincoln, NE) and 680LT goat anti-mouse (1:10,000) (926–68021; LI-COR Biosciences).

## Acknowledgements

We thank Ashley Yu for assistance with cloning, and all the members of the Emr laboratory for helpful discussions.

## Additional information

### Funding

| Funder | Grant reference number | Author |
| --- | --- | --- |
| Cornell University Research Grant | CU3704 | Scott D Emr |
| National Institutes of Health | R35GM136258 | J Christopher Fromme |

The funders had no role in study design, data collection and interpretation, or the decision to submit the work for publication.

### Author contributions

Richa Sardana, Conceptualization, Data curation, Formal analysis, Investigation, Methodology, Resources, Validation, Visualization, Writing – original draft, Writing – review and editing; Carolyn M Highland, Investigation, Visualization, Writing – review and editing; Beth E Straight, Christopher F Chavez, Investigation; J Christopher Fromme, Writing – review and editing; Scott D Emr, Conceptualization, Funding acquisition, Supervision, Writing – review and editing

### Author ORCIDs

Richa Sardana http://orcid.org/0000-0001-5861-9648
Carolyn M Highland http://orcid.org/0000-0003-4029-0113
J Christopher Fromme http://orcid.org/0000-0002-8837-0473
Scott D Emr http://orcid.org/0000-0002-5408-6781

**Decision letter and Author response**
Decision letter https://doi.org/10.7554/eLife.70774.sa1
Author response https://doi.org/10.7554/eLife.70774.sa2

---

## Additional files

### Supplementary files
• Supplementary file 1. List of strains and plasmids used in this study.

• Transparent reporting form

### Data availability
All data generated or analyzed during this study are included in the manuscript and supporting files. Source data files have been provided for Figures 1, 2, 3, Fig 2-S2, Fig 3-S1.

The following previously published datasets were used:

| Author(s) | Year | Dataset title | Dataset URL | Database and Identifier |
|---|---|---|---|---|
| Usaj M, Tan Y, Wang W, VanderSluis B, Zou A, Myers CL, Costanzo M, Andrews B, Boone C | 2017 | TheCellMap.org: A Web-Accessible Database for Visualizing and Mining the Global Yeast Genetic Interaction Network | https://thecellmap.org/ | thecellmap, 10.1534/g3.117.040220 |

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
