## [Editor Report]

The authors took the critiques seriously and responded with substantial changes and enhancements. While we are still not fully convinced by the interpretations, the authors have done a thorough, rigorous piece of work that makes a valuable contribution about a topic of broad interest.

---

## [Decision Letter]

**Decision letter after peer review:**

Thank you for submitting your article "Golgi membrane protein Erd1 is essential for recycling of Golgi glycosyltransferases" for consideration by *eLife*. Your article has been reviewed by 3 peer reviewers, including Benjamin S Glick as Reviewing Editor and Reviewer #1, and the evaluation has been overseen by Vivek Malhotra as the Senior Editor. The following individual involved in review of your submission has agreed to reveal their identity: Alberto Luini (Reviewer #2).

Essential revisions:

The reason for these revisions is that the evidence about the role of Erd1 could be consistent with mechanisms different from the one presented in the manuscript. In the spirit of constructive feedback, the authors are encouraged to dig a bit deeper and potentially reconsider their conclusions.

1) Perform a simultaneous kinetic analysis of Erd1 and Vps74 to determine the extent of temporal overlap between these two proteins during maturation. This result will help to clarify the functional relationship between Erd1 and Vps74, and may point to a more complex model along the lines of the suggestion from Reviewer #2.

A related issue is that Erd1 seems to resemble glycosyltransferases in being a cargo of Vps74-mediated recycling. This point should be kept in mind when considering whether the Erd1-Vps74 interaction has a richer meaning.

2) Provide evidence for direct interactions between Erd1 and glycosyltransferases. Such data are important to support a model in which Erd1 plays a receptor-like role in recycling.

3) Carefully evaluate the earlier work implicating Erd1 in fungal phosphate transport, especially in light of the statement by Reviewer #3 that the entire Erd1 protein is an EXS domain. A key paper here is reference 46 in the manuscript.

Proteins known to maintain the luminal ionic environment of the Golgi, such as Pmr1, are also required for normal glycosylation. It will be informative if the authors can present arguments that Erd1 is in a different category.

4) Cite and discuss the earlier paper by Okamoto et al.

*Reviewer #1:*

Glycosylation in the Golgi is carried out by resident enzymes that recycle within the organelle while secretory proteins move forward in the maturing cisternae. The mechanism of Golgi enzyme recycling is incompletely understood, but it seems to involve retrograde COPI coated vesicles. In yeast, a subset of Golgi enzymes interact with the peripheral membrane protein Vps74, which is thought to serve as an adaptor for capturing Golgi enzymes in COPI vesicles. The current manuscript identifies the integral membrane protein Erd1 as a facilitator of Golgi enzyme recycling. Evidence is presented that Erd1 interacts with Golgi enzymes as well as Vps74, and that Erd1 is needed for normal Golgi enzyme localization. Therefore, Erd1 is an important new player in the maintenance of Golgi organization.

This study is of high quality. The data are generally strong, the experiments are logical, and multiple complementary approaches are used to answer the questions. A persuasive case is made that Erd1 plays a role in recycling Golgi enzymes.

Less persuasive is the interpretation that Erd1 directly activates Vps74 to drive recycling. Much of the evidence is circumstantial--e.g., loss of Erd1 and loss of Vps74 confer similar phenotypes. Skepticism about the authors' interpretation centers around timing during the maturation process:

– Two of the authors (Highland and Fromme) recently published an analysis of PI4P dynamics, and they concluded that PI4P is a hallmark of the late Golgi, sharing "little overlap with markers of less mature Golgi compartments".

– Vps74 has been found to require PI4P for its recruitment, so Vps74 would be expected to reside at the late Golgi. Indeed, even though Vps74 peaks earlier than the late Golgi marker Sec7, it seems to arrive at about the same time as Sec7.

– By contrast, Erd1 clearly overlaps with early Golgi enzymes, which have mostly departed by the time that Sec7 arrives. It is therefore unclear whether Erd1 and Vps74 overlap for long enough to engage in a direct physical interaction that would drive Golgi enzyme recycling.

The major recommendation, intended to address the concern raised above is to visualize the maturation kinetics of Erd1 and Vps74 simultaneously.

*Reviewer #2:*

Based on bioinformatic, biochemical, morphological and genetic data, the authors propose a model of the mechanism of action of ERD1 according to which enzymes belonging to a selected group of glycosyltransferases are retained in the Golgi by recycling via COPI vesicles. According to this model the incorporation of enzymes in vesicles requires a sequential interaction of these enzymes first with ERD1 in the cis golgi and then with GOLPH3 in the medial trans Golgi. The first interaction has, in the proposed model, a preparatory or facilitating role for the second interaction, with GOLPH3, which mediates entry into COPI -dependent retrograde transport vesicles, and therefore the recycling, of the enzymes.

This is an interesting study in the rapidly developing area of the recycling mechanisms of glycosylation enzymes in Golgi. However, there are some weaknesses, which are outlined below.

The biochemical data on the binding between ERD1, VPS74 and Golgi enzymes are incomplete and should be completed with the demonstration that ERD1 and client enzymes interact directly, by means of the Y2H technique. This interaction is so far only demonstrated by co-precipitation of ERD1 with GOLPH3 and the enzymes, but these experiments do not exclude that ERD1 which interacts directly with GOLPH3, might interact indirectly with the enzymes, through its direct interaction with GOLPH3. Since the interaction between ERD1 and the client enzymes is at the heart of this study, it should be demonstrated with dedicated experiments.

On the technical plane, in the experiment in Figure 3C the efficiencies of co-ip in WT cells and in deltaERD1, once corrected for the inputs, do not seem very different. It would be useful to show inputs and IPs in the very same blot and to provide a quantification graph.

The localization of ERD1, GOLPH3 and enzymes are interpreted by the authors in the context of the ERD1/GOLPH3 synergy model briefly outlined above, and presented in figure 4. However, not all of the data are easily reconcilable with this model.

The main difficulty is that both ERD1 and its client enzymes appear and recycle in the cis cisterna while GOLPH3 resides in the medial and partly in the trans cisternae, but not in the cis cisterna. Moreover, the scheme presented by the authors in Figure 4 indicates that the recycling of the cis enzymes (eg, Mann9) begins before the appearance of GOLPH3 in the cis cisterna. This is in principle not compatible with the model, unless the authors propose that non-detectable traces of GOLPH3 might be present in the cis cisterna and activate the recycling mechanism there. If the authors show that this is the case, it might be acceptable to present their model in this study. However, given that our understanding of Golgi enzyme recycling is actually in its infancy, the underlying mechanism might be more complex. For example, ERD1 could mediate the recycling of client enzymes in the cis cisterna in synergy with a hitherto unknown adapter other than GOLPH3 (perhaps COPI itself?), and might also mediate the recycling of enzymes in synergy with GOLPH3 in the medial cisterna. This is possible because according to the scheme in figure 4 ERD1 shows a peak in the cis cisterna but is clearly present also in the medial cisterna. Furthermore, all ERD1-dependent enzymes should also be indirectly dependent on GOLPH3 since GOLPH3 depletion causes ERD1 loss and lysosomal degradation. ERD1 might thus be an adapter capable of acting at various levels of the transport system: in Golgi to ER recycling and in glycoenzyme recycling at both the cis and the medial cisterna. This model can explain the data probably better than that presented by the authors in Figure 4, but is more complicated and requires some more assumptions. It is up to the authors to decide whether they want to consider this hypothesis as an alternative model.

A question that would need to be addressed in this context is whether all the enzymes found by the authors to depend on ERD1 are also GOLPH3 customers and in particular whether they all contain the recognition motif (F / L) – (L / I / V ) -xx- (R / K) for GOLPH3 described by Banfield.

My recommendations to the authors have already been outlined above. In short, I think that the authors should:

Characterize the interaction between GOLPH3 and client enzymes by Y2H.

Discuss the weaknesses of their model as presented in figure 4 and, if they wish, mention the possibility of a more complex mechanism,

Clarify whether all ERD1 client enzymes contain the motif (F / L) – (L / I / V) -x-x- (R / K),

Discuss the earlier paper by Okamoto et al., (2008) that is not cited in this study.

In my opinion the authors should also discuss two recent papers (Rizzo et al; Parashurama et al., both in EMBO J), that are based on the concepts of recycling adapter and retention adapter and propose a model for the localization of GOLPH3-dependent enzymes in Golgi compartments in the context of cisternal maturation.

*Reviewer #3:*

The authors investigate the role of the yeast protein Erd1 in Golgi-dependent glycosylation. They conclude that Erd1 acts with Vps74, a known retention factor for Golgi glycosyltransferases, to direct the recycling of these enzymes in COPI-coated vesicles in the Golgi stack. The data are well presented and quantified, and the paper is clearly written. Addressing the role of Erd1, and insight into how glycosyltransferases are retained in the Golgi, are both interesting questions, but the authors data do not preclude alternative interpretations, and one or two aspects require resolution. These issues are summarized below:

1) A role for Erd1 in acting as a coreceptor with Vps74 to recycle glycosyltransferases is interesting but also raises some questions. Firstly, Erd1 is only found in fungi and not in metazoans, whereas Vps74 has metazoan orthologs that are known to play a role in glycosyltransferase recycling raising the question of why Erd1 is only needed in yeast. Secondly, there is published evidence for Erd1 acting as a channel/transporter for the movement of phosphate out of the Golgi lumen, and indeed the entire protein comprises a domain (the EXS domain) that is present in known phosphate transporters in plants and metazoans. Thus, careful dissection is required to determine if the effects seen are direct via an interaction with glycosylation enzymes, or if they are an indirect consequence of a perturbation of the Golgi lumen due to accumulation of inorganic phosphate. The fact that Erd1 was originally identified as having a defect in the retrieval of soluble ER residents from the Golgi suggests that the Golgi lumen may well be perturbed, possibly by a change in pH or cation content, and it is known that alterations in these features of the Golgi can affect glycosylation.

2) Perhaps the most striking finding is that Erd1 co-precipitates with Vps74 and that the presence of Erd1 is required for Vps74 to efficiently co-precipitate with the glycosyltransfereases that it is known to bind. However, the authors also show that the glycosyltransferases are destabilized by the loss of Erd1, and so the loss of apparent interaction may simply reflect the fact that there is less protein present to co-precipitate. Secondly, the authors show that Erd1 and Vps74 do not substantially co-localize, and so any tripartite complex would have to reflect a small sub-population of the proteins that briefly come together later in the Golgi stack during formation of COPI coated vesicles that are to be recycled. Finally, Vps74 is known to bind to COPI, and so if Erd1 also bound to COPI, then Vps74 and Erd1 may co-precipitate because they are held together via COPI, with Vps74 then bringing some glycosyltransferases into the complex.

3) Some aspects of the data may need resolving. Firstly, the authors provide clear images showing degradation of Och1p-GFP and Kre2-GFP in the vacuole in the absence of Erd1 (Figure 1F). However, the immunoblots in Figure 3D indicate that the levels of the intact proteins are unchanged in the absence of Erd1 which suggests that they are not destabilized and degraded. Secondly, the authors use the split ubiquitin system to provide evidence for an interaction between Erd1 and Vps74. As a control they remove the "cytoplasmic-tail" from Erd1, but they do not state how many residues were removed. The structural prediction for the EXS domain in Pfam suggests that the last membrane spanning helix of Erd1 would be very close to the C-terminus (Pfam entry PF03124), and the location of the truncation is not tested. Finally, the authors show some nice live cell imaging data to follow Golgi maturation. However, they do not directly compare Vps74 and Erd1. Such a comparison would be very helpful, especially as it seems from the other graphs that Erd1 is significantly depleted from the maturing cisterna before the time when there are substantial amounts of Vps74 present.

The prior publications on Erd1, the absence of an orthologue in mammals even though they have a Vps74 ortholog, and the potential role of COPI as a bridge between the two proteins, really necessitate a much more in depth and substantial analysis for a broad readership journal such as *eLife*. Ideally, in vitro reconstitution of binding with purified proteins would resolve many issues, but I appreciate that this may be technically challenging. Below I have suggested some things that could be done to strengthen the paper's conclusions, and at the very least these may be helpful to the authors to consider before resubmitting elsewhere:

1) Examination of the effect of other gene deletions that affect the ionic content of the Golgi such as deletion of Gdt1, Pmr1 or Stv1, on Och1p-GFP and Ktr2p-GFP levels and on glycosylation.

2) Resolution of the apparent contradiction of the effects of Erd1 deletion on Och1 and Ktr2 by microscopy and blotting assays. If available, antibodies to the endogenous proteins could be used to test their levels in wild type and mutants.

3) Does the mutation in the COPI binding site of Vps74 affect its Golgi localization? If not, the authors should check if this prevents the co-ip with Erd1.

4) It would be very valuable to add videos and graphs to follow the Golgi localization of Erd1 vs Vps74 to better reveal their spatial relationship over time.

5) Substantial new insight would be provided by determining what part of at least of the Erd1-dependent glycosyltransferases interacts with Erd1. This could be addressed by making chimeric proteins that contain only either the cytoplasmic tail, or the TMD, or the lumenal domain of a Erd1-dependent glycosyltransferase in the context of an Erd1-indepenent glycosyltransferase. The localization and co-ip of these chimeras could be then be tested.

6) The authors argue that the cytoplasmic tail of Erd1 interacts with Vps74. This could be tested biochemically as has been done for the tails of glycosyltransferases. If the authors keep the split-ubiquitin experiments they should confirm that the constructs are localized to the Golgi.

7) Are their glycosyltransferases that do no rely on Vps74? It would be useful to test if these are affected by loss of Erd1.

---

## [Author Response]

Essential revisions:The reason for these revisions is that the evidence about the role of Erd1 could be consistent with mechanisms different from the one presented in the manuscript. In the spirit of constructive feedback, the authors are encouraged to dig a bit deeper and potentially reconsider their conclusions.1) Perform a simultaneous kinetic analysis of Erd1 and Vps74 to determine the extent of temporal overlap between these two proteins during maturation. This result will help to clarify the functional relationship between Erd1 and Vps74, and may point to a more complex model along the lines of the suggestion from Reviewer #2.

As per the reviewers’ request, we have now performed simultaneous kinetic analysis of Erd1 and Vps74 and included the data in Figure 4G. Kinetic analysis of Erd1 and Vps74 suggests a temporal overlap between the two, consistent with our model that arrival of Vps74 coincides with the packaging and departure of the cargos in COPI vesicles. Erd1 kinetics resemble earlyGolgi proteins, some of which have been previously reported to be Vps74 clients and have a similar overlap time with Vps74 as Erd1. The temporal overlap is also consistent with our kinetic analysis of Erd1 and Vps74 with early (Mnn9), medial (Gea2) and late (Sec7) Golgi markers (see Figure 4 A-F and Figure 4-supplement 1).

A related issue is that Erd1 seems to resemble glycosyltransferases in being a cargo of Vps74-mediated recycling. This point should be kept in mind when considering whether the Erd1-Vps74 interaction has a richer meaning.

We agree with the reviewers, that Erd1 indeed acts as a mediator and a cargo of Vps74dependent recycling is quite intriguing. By chaperoning the glycosyltransferase cargos throughout the recycling process, Erd1 appears to play a unique role in ensuring productive recycling of the cargos. To our knowledge, Erd1 is also the first example of a multi-pass membrane protein recycled via this pathway.

2) Provide evidence for direct interactions between Erd1 and glycosyltransferases. Such data are important to support a model in which Erd1 plays a receptor-like role in recycling.

As suggested by reviewer 2, we first tested the interaction between full length Erd1 and the glycosyltransferases using the split-Ub Y2H assay but did not observe a robust interaction. The nature of the Y2H assay requires the tag to be present on the cytosolic facing end of the proteins being tested for interaction. All glycosyltransferases are Type II single TM proteins, with the N terminus in the cytosol, and the C terminus in the lumen. To make sure the tagged constructs localized as expected, we tested the localization of N- and C-terminally GFP tagged glycosyltransferases, Kre2 and Och1. As shown in Author response image 1, while C-terminal tagged Kre2-GFP exhibits a punctate Golgi localization (right), N-terminal tagged GFP-Kre2 (left) is significantly retained in the ER (likely by interfering with COPII packaging). These observations clarified why the Y2H assay was unable to score the interaction between Erd1 and the glycosyltransferases.

**Author response image 1. sa2fig1:** 

We also attempted to purify GST tagged full length Erd1 from yeast cells for in vitro binding experiments but were unable to obtain enough protein needed for the analysis. Finally, as an alternative strategy, we employed bimolecular fluorescence complementation (BiFC) to test interaction between Erd1 and glycosyltransferases with tags fused to the termini facing the lumen. This assay has been employed to score interactions between Vps74-Vps74 and Vps74Sac1 previously (Cai et al., JCB 2014, Wood et al., MBoC 2012). We observed reconstitution of fluorescence signal in the BiFC assay in strains expressing Och1-mNG^1-10^ and Erd1-mNG^11^. As expected, the BiFC signal co-localized with the cis-Golgi marker Mnn9-mCherry. We have included this data in Figure 3A. While we cannot rule out the possibility of an indirect interaction, these results in combination with other data presented in the manuscript, make a compelling case for a close interaction between Erd1 and its client glycosyltransferases.

3) Carefully evaluate the earlier work implicating Erd1 in fungal phosphate transport, especially in light of the statement by Reviewer #3 that the entire Erd1 protein is an EXS domain. A key paper here is reference 46 in the manuscript.Proteins known to maintain the luminal ionic environment of the Golgi, such as Pmr1, are also required for normal glycosylation. It will be informative if the authors can present arguments that Erd1 is in a different category.

Erd1 has indeed been proposed to play a role in phosphate transport owing to the conserved EXS domain, and interactions reported by Snyder et al., 2017. We completely agree with the reviewers that the ionic environment of the Golgi is expected to affect normal glycosylation. We have now included multiple controls, that demonstrate that our observations on the role in Golgi glycosyltransferase recycling is distinct from just an effect on the ionic environment.

As requested by the reviewers, we compared the effects of Erd1 and Pmr1 (Golgi Ca^2+^ and Mn^2+^ transporter) on glycosylation and glycosyltransferase recycling. We observed that both erd1∆ and pmr1∆ mutants exhibit defects in glycosylation of Gas1 reporter (see Figure 2- supplement 2A). However, unlike erd1∆ mutant, the pmr1∆ mutant shows no defect in the localization and recycling of Kre2 (see Figure 2- supplement 2D). If the primary defect is an ionic imbalance, then supplementing the media with the key ion will be expected to rescue the observed glycosylation defects. Synder et al., had reported that erd1∆ mutant requires 1.5fold higher phosphate concentration in the media for 50% maximal growth as compared to wild type, and that Erd1 likely limits the export of Pi from wild type cells via exocytosis. However, supplementing the media with higher concentration of Pi (even at concentrations needed to support the growth of erd1∆ pho∆-5 mutant reported in Synder et al.,) did not rescue the glycosylation defects observed in erd1∆ (see Figure 2- supplement 2B). On the other hand, supplementing the media with Ca^2+^ (complete rescue) or Mn^2+^ (partial rescue) suppressed the glycosylation defect observed in pmr1∆ mutant (see Figure 2- supplement 2C). Therefore, while it is possible that Erd1 plays a role in phosphate homeostasis at the early Golgi, our data support an additional role of Erd1 (distinct from other ion transporters at the Golgi) in also being required for glycosyltransferase recycling.

4) Cite and discuss the earlier paper by Okamoto et al.

We thank the reviewers for pointing this out and apologize for the oversight. We have now cited the paper and included it in the discussion.

Reviewer #1:Glycosylation in the Golgi is carried out by resident enzymes that recycle within the organelle while secretory proteins move forward in the maturing cisternae. The mechanism of Golgi enzyme recycling is incompletely understood, but it seems to involve retrograde COPI coated vesicles. In yeast, a subset of Golgi enzymes interact with the peripheral membrane protein Vps74, which is thought to serve as an adaptor for capturing Golgi enzymes in COPI vesicles. The current manuscript identifies the integral membrane protein Erd1 as a facilitator of Golgi enzyme recycling. Evidence is presented that Erd1 interacts with Golgi enzymes as well as Vps74, and that Erd1 is needed for normal Golgi enzyme localization. Therefore, Erd1 is an important new player in the maintenance of Golgi organization.This study is of high quality. The data are generally strong, the experiments are logical, and multiple complementary approaches are used to answer the questions. A persuasive case is made that Erd1 plays a role in recycling Golgi enzymes.Less persuasive is the interpretation that Erd1 directly activates Vps74 to drive recycling. Much of the evidence is circumstantial--e.g., loss of Erd1 and loss of Vps74 confer similar phenotypes. Skepticism about the authors' interpretation centers around timing during the maturation process:– Two of the authors (Highland and Fromme) recently published an analysis of PI4P dynamics, and they concluded that PI4P is a hallmark of the late Golgi, sharing "little overlap with markers of less mature Golgi compartments".– Vps74 has been found to require PI4P for its recruitment, so Vps74 would be expected to reside at the late Golgi. Indeed, even though Vps74 peaks earlier than the late Golgi marker Sec7, it seems to arrive at about the same time as Sec7.– By contrast, Erd1 clearly overlaps with early Golgi enzymes, which have mostly departed by the time that Sec7 arrives. It is therefore unclear whether Erd1 and Vps74 overlap for long enough to engage in a direct physical interaction that would drive Golgi enzyme recycling.

We thank the reviewer for recognizing our work and the importance of our findings.

We find it quite interesting how Erd1, an integral membrane protein that localizes with its client early Golgi enzymes, and Vps74, a peripheral membrane protein that is recruited to medial Golgi compartments can cooperatively facilitate recycling. As reported in Highland et al., PI4P is indeed a hallmark of the late Golgi compartments. Additionally, previous work from Wood et al., has shown the important of PI4P binding for stable association of Vps74 with the Golgi, and the co-localization of Vps74 with medial Golgi markers. Our kinetic analysis of Vps74 association with the Golgi in the context of Golgi maturation agree with these previous findings and suggest more complex dynamics than just PI4P association. We find that Vps74 arrives at the Golgi after the medial Golgi marker, Gea2 when the PI4P levels are low, but perhaps enough to recruit Vps74. Vps74 peaks and leaves before peak Sec7 or PI4P signal. These observations suggest that in addition to PI4P, protein-protein interactions likely also play a role in regulating Vps74 recruitment and departure at the medial-late Golgi transition. Simultaneous analysis of Erd1 and Vps74 indicates a temporal overlap, and arrival of Vps74 marks the beginning of departure of Erd1.

The major recommendation, intended to address the concern raised above is to visualize the maturation kinetics of Erd1 and Vps74 simultaneously.

As suggested by the reviewer, we have now included simultaneous kinetic analysis of Erd1 and Vps74. This data is included in Figure 4G. As mentioned above, these results are consistent with our model.

Reviewer #2:Based on bioinformatic, biochemical, morphological and genetic data, the authors propose a model of the mechanism of action of ERD1 according to which enzymes belonging to a selected group of glycosyltransferases are retained in the Golgi by recycling via COPI vesicles. According to this model the incorporation of enzymes in vesicles requires a sequential interaction of these enzymes first with ERD1 in the cis golgi and then with GOLPH3 in the medial trans Golgi. The first interaction has, in the proposed model, a preparatory or facilitating role for the second interaction, with GOLPH3, which mediates entry into COPI -dependent retrograde transport vesicles, and therefore the recycling, of the enzymes.This is an interesting study in the rapidly developing area of the recycling mechanisms of glycosylation enzymes in Golgi. However, there are some weaknesses, which are outlined below.The biochemical data on the binding between ERD1, VPS74 and Golgi enzymes are incomplete and should be completed with the demonstration that ERD1 and client enzymes interact directly, by means of the Y2H technique. This interaction is so far only demonstrated by co-precipitation of ERD1 with GOLPH3 and the enzymes, but these experiments do not exclude that ERD1 which interacts directly with GOLPH3, might interact indirectly with the enzymes, through its direct interaction with GOLPH3. Since the interaction between ERD1 and the client enzymes is at the heart of this study, it should be demonstrated with dedicated experiments.On the technical plane, in the experiment in Figure 3C the efficiencies of co-ip in WT cells and in deltaERD1, once corrected for the inputs, do not seem very different. It would be useful to show inputs and Ips in the very same blot and to provide a quantification graph.

We thank the reviewer for the constructive suggestions.

As mentioned in the response to essential revisions above, we were unable to utilize the Y2H assay to score interaction between Erd1 and glycosyltransferases due to interference of the Nterminal Y2H tag with the Golgi localization of glycosyltransferases. By tagging the uminal ends, we were able to successfully employ a BiFC based approach to monitor interaction between Erd1 and Och1, as well as that between Erd1 and Vps74. These data are included in Figure 3A. We have also included the inputs and Ips from WT and erd1∆ mutant on the same blot and added a quantitation graph from multiple repeats of the experiment. These data are included in Figure 3D, 3E, Figure 3- supplement 1E.

The localization of ERD1, GOLPH3 and enzymes are interpreted by the authors in the context of the ERD1/GOLPH3 synergy model briefly outlined above, and presented in figure 4. However, not all of the data are easily reconcilable with this model.The main difficulty is that both ERD1 and its client enzymes appear and recycle in the cis cisterna while GOLPH3 resides in the medial and partly in the trans cisternae, but not in the cis cisterna. Moreover, the scheme presented by the authors in Figure 4 indicates that the recycling of the cis enzymes (eg, Mann9) begins before the appearance of GOLPH3 in the cis cisterna. This is in principle not compatible with the model, unless the authors propose that non-detectable traces of GOLPH3 might be present in the cis cisterna and activate the recycling mechanism there. If the authors show that this is the case, it might be acceptable to present their model in this study. However, given that our understanding of Golgi enzyme recycling is actually in its infancy, the underlying mechanism might be more complex. For example, ERD1 could mediate the recycling of client enzymes in the cis cisterna in synergy with a hitherto unknown adapter other than GOLPH3 (perhaps COPI itself?), and might also mediate the recycling of enzymes in synergy with GOLPH3 in the medial cisterna. This is possible because according to the scheme in figure 4 ERD1 shows a peak in the cis cisterna but is clearly present also in the medial cisterna. Furthermore, all ERD1-dependent enzymes should also be indirectly dependent on GOLPH3 since GOLPH3 depletion causes ERD1 loss and lysosomal degradation. ERD1 might thus be an adapter capable of acting at various levels of the transport system: in Golgi to ER recycling and in glycoenzyme recycling at both the cis and the medial cisterna. This model can explain the data probably better than that presented by the authors in Figure 4, but is more complicated and requires some more assumptions. It is up to the authors to decide whether they want to consider this hypothesis as an alternative model.A question that would need to be addressed in this context is whether all the enzymes found by the authors to depend on ERD1 are also GOLPH3 customers and in particular whether they all contain the recognition motif (F / L) – (L / I / V ) -xx- (R / K) for GOLPH3 described by Banfield.

While we present a simpler model, we agree with the reviewer that it is possible that the mechanism of protein recycling in the Golgi may be more complex for different cargos- with Erd1 assisting in Vps74-independent (for Golgi to ER recycling) and Vps74-dependent (for intra- Golgi recycling) steps. Indeed, Erd1 was identified in a screen for mutants defective in recycling of ER residents. Exploring these roles of Erd1 would require a more systematic approach in the future.

Using simultaneous kinetic analysis, we find that Erd1 and Vps74 do overlap (figure 4G). This interaction can also be captured by the BiFC and immunoprecipitation (figure 3A, 3B). Erd1 kinetics resemble early-Golgi proteins, some of which have been previously reported to be Vps74 clients and have a similar overlap time with Vps74 as Erd1. We think that these observations support a synergy model, but we agree with the reviewer about the possibility of a dual step role for Erd1.

As the reviewer correctly noted, since Erd1 itself requires Vps74 for its recycling, it is difficult to tease apart Erd1-only from Vps74 dependent clients. Indeed, all the glycosyltransferases that we tested and found to be dependent on Erd1, were also dependent on Vps74 for their recycling. This subset of glycoenzymes do appear to contain (F/L)-(I/L/V)-x-x-(R/K) sequence in their cytosolic tails previously proposed to bind Vps74 (figure 2- supplement 2E).

My recommendations to the authors have already been outlined above. In short, I think that the authors should:Characterize the interaction between GOLPH3 and client enzymes by Y2H.Discuss the weaknesses of their model as presented in figure 4 and, if they wish, mention the possibility of a more complex mechanism,Clarify whether all ERD1 client enzymes contain the motif (F / L) – (L / I / V) -x-x- (R / K),Discuss the earlier paper by Okamoto et al., (2008) that is not cited in this study.In my opinion the authors should also discuss two recent papers (Rizzo et al; Parashurama et al., both in EMBO J), that are based on the concepts of recycling adapter and retention adapter and propose a model for the localization of GOLPH3-dependent enzymes in Golgi compartments in the context of cisternal maturation.

We thank the reviewer for the feedback. As listed above, we have addressed all the reviewer comments. We have also discussed the work published by Okamoto et al., Rizzo et al., and Pothukuchi…Parashuraman et al., in the text.

Reviewer #3:The authors investigate the role of the yeast protein Erd1 in Golgi-dependent glycosylation. They conclude that Erd1 acts with Vps74, a known retention factor for Golgi glycosyltransferases, to direct the recycling of these enzymes in COPI-coated vesicles in the Golgi stack. The data are well presented and quantified, and the paper is clearly written. Addressing the role of Erd1, and insight into how glycosyltransferases are retained in the Golgi, are both interesting questions, but the authors data do not preclude alternative interpretations, and one or two aspects require resolution. These issues are summarized below:1) A role for Erd1 in acting as a coreceptor with Vps74 to recycle glycosyltransferases is interesting but also raises some questions. Firstly, Erd1 is only found in fungi and not in metazoans, whereas Vps74 has metazoan orthologs that are known to play a role in glycosyltransferase recycling raising the question of why Erd1 is only needed in yeast. Secondly, there is published evidence for Erd1 acting as a channel/transporter for the movement of phosphate out of the Golgi lumen, and indeed the entire protein comprises a domain (the EXS domain) that is present in known phosphate transporters in plants and metazoans. Thus, careful dissection is required to determine if the effects seen are direct via an interaction with glycosylation enzymes, or if they are an indirect consequence of a perturbation of the Golgi lumen due to accumulation of inorganic phosphate. The fact that Erd1 was originally identified as having a defect in the retrieval of soluble ER residents from the Golgi suggests that the Golgi lumen may well be perturbed, possibly by a change in pH or cation content, and it is known that alterations in these features of the Golgi can affect glycosylation.2) Perhaps the most striking finding is that Erd1 co-precipitates with Vps74 and that the presence of Erd1 is required for Vps74 to efficiently co-precipitate with the glycosyltransfereases that it is known to bind. However, the authors also show that the glycosyltransferases are destabilized by the loss of Erd1, and so the loss of apparent interaction may simply reflect the fact that there is less protein present to co-precipitate. Secondly, the authors show that Erd1 and Vps74 do not substantially co-localize, and so any tripartite complex would have to reflect a small sub-population of the proteins that briefly come together later in the Golgi stack during formation of COPI coated vesicles that are to be recycled. Finally, Vps74 is known to bind to COPI, and so if Erd1 also bound to COPI, then Vps74 and Erd1 may co-precipitate because they are held together via COPI, with Vps74 then bringing some glycosyltransferases into the complex.3) Some aspects of the data may need resolving. Firstly, the authors provide clear images showing degradation of Och1p-GFP and Kre2-GFP in the vacuole in the absence of Erd1 (Figure 1F). However, the immunoblots in Figure 3D indicate that the levels of the intact proteins are unchanged in the absence of Erd1 which suggests that they are not destabilized and degraded. Secondly, the authors use the split ubiquitin system to provide evidence for an interaction between Erd1 and Vps74. As a control they remove the "cytoplasmic-tail" from Erd1, but they do not state how many residues were removed. The structural prediction for the EXS domain in Pfam suggests that the last membrane spanning helix of Erd1 would be very close to the C-terminus (Pfam entry PF03124), and the location of the truncation is not tested. Finally, the authors show some nice live cell imaging data to follow Golgi maturation. However, they do not directly compare Vps74 and Erd1. Such a comparison would be very helpful, especially as it seems from the other graphs that Erd1 is significantly depleted from the maturing cisterna before the time when there are substantial amounts of Vps74 present.

We thank the reviewer for the comments. We acknowledge that the role of Erd1 in phosphate transport is likely an important function. However, as mentioned above in our response to essential revision, we have now included data to show the effect on protein recycling upon the loss of Erd1 cannot be reconciled by just changes in the Golgi ionic environment, and this effect is distinct from that due to the loss of Pmr1 (Figure 2- supplement 2). Our observations also suggest that like Erd1 in yeast, other transmembrane proteins may play similar roles in GOLPH3 dependent recycling in metazoans. We were also surprised to observe that despite the recycling defect observed in erd1∆ and vps74∆ mutants, we saw a modest reduction in steady state levels of tagged Kre2 or Och1 on western blots. This may be reflective of upregulated biosynthesis or slower degradation of these constructs in the vacuolar lumen. We also confirmed that tagged Och1 can complement the temperature sensitivity of och1∆ mutant and is functional (Figure 4- supplement 1D). Reduced co-immunoprecipitation of Vps74 and Och1/Kre2 in the absence of Erd1 may be exacerbated by the fact that a significant fraction of Och1/Kre2 are sequestered away from the Golgi. However, this is in part mediated because of a defect in productive recycling complex formation required to maintain these enzymes at the Golgi, highlighting the importance of Erd1’s role in the pathway.

We have now included a simultaneous kinetic analysis of Vps74 and Erd1 and show that there is overlap between the two (figure 4G). Additionally, using split-fluorescent protein complementation to monitor protein-protein interactions, we show that Erd1 interacts with wild type Vps74, as well as Vps74 R6-8A COPI binding mutant, suggesting the interaction between Erd1 and Vps74 is not bridged via COPI binding (figure 3A).

The C-terminal truncation of Erd1 employed in the Y2H assay, was initially generated based on the original truncation mutation identified in the ‘erd’ screen by Hardwick et al., along with the cytosolic region prediction on the Uniprot database and lacks the last 100 aa. Based on structural modeling of Erd1 using AlphaFold, we agree that truncation likely disrupts the intramembrane helices of the EXS domain. We subsequently also generated truncations of Cterminal 10 and 50 aa to test Y2H interaction with Vps74, but they all affected protein stability. It is unclear if they are degraded because of loss of interaction with Vps74 or structural instability and cannot be conclusively used to inform of the interaction region with Vps74 and have not been included in the revised manuscript. How Erd1 interacts with Vps74 will require more in-depth analysis in the future.

The prior publications on Erd1, the absence of an orthologue in mammals even though they have a Vps74 ortholog, and the potential role of COPI as a bridge between the two proteins, really necessitate a much more in depth and substantial analysis for a broad readership journal such as eLife. Ideally, in vitro reconstitution of binding with purified proteins would resolve many issues, but I appreciate that this may be technically challenging. Below I have suggested some things that could be done to strengthen the paper's conclusions, and at the very least these may be helpful to the authors to consider before resubmitting elsewhere:1) Examination of the effect of other gene deletions that affect the ionic content of the Golgi such as deletion of Gdt1, Pmr1 or Stv1, on Och1p-GFP and Ktr2p-GFP levels and on glycosylation.2) Resolution of the apparent contradiction of the effects of Erd1 deletion on Och1 and Ktr2 by microscopy and blotting assays. If available, antibodies to the endogenous proteins could be used to test their levels in wild type and mutants.3) Does the mutation in the COPI binding site of Vps74 affect its Golgi localization? If not, the authors should check if this prevents the co-ip with Erd1.4) It would be very valuable to add videos and graphs to follow the Golgi localization of Erd1 vs Vps74 to better reveal their spatial relationship over time.

Please see our detailed response above that addresses these points.

5) Substantial new insight would be provided by determining what part of at least of the Erd1-dependent glycosyltransferases interacts with Erd1. This could be addressed by making chimeric proteins that contain only either the cytoplasmic tail, or the TMD, or the lumenal domain of a Erd1-dependent glycosyltransferase in the context of an Erd1-indepenent glycosyltransferase. The localization and co-ip of these chimeras could be then be tested.6) The authors argue that the cytoplasmic tail of Erd1 interacts with Vps74. This could be tested biochemically as has been done for the tails of glycosyltransferases. If the authors keep the split-ubiquitin experiments they should confirm that the constructs are localized to the Golgi.7) Are their glycosyltransferases that do no rely on Vps74? It would be useful to test if these are affected by loss of Erd1.

We have attempted to generate chimeric proteins by swapping the cytoplasmic or transmembrane domain of an Erd1-dependent and independent glycosyltransferase. However, the chimeric protein was unstable and degraded even in wild type cells. While we cannot address the comments about binding determinants with our current toolset, this will be the focus our investigations in the future. All the glycosyltransferases that we have tested that rely on Erd1, also rely on Vps74, so we are currently unable to test the effect of Erd1 or Vps74 alone.